# Synthesis, Characterization, and Catalytic Application of Palladium Complexes Containing Indolyl-NNN-Type Ligands

**DOI:** 10.3390/molecules26154426

**Published:** 2021-07-22

**Authors:** Pang-Chia Lo, Chun-Wei Yang, Wen-Kai Wu, Chi-Tien Chen

**Affiliations:** Department of Chemistry, National Chung Hsing University, Taichung 402, Taiwan; k101312001@gmail.com (P.-C.L.); g104051082@mail.nchu.edu.tw (C.-W.Y.); g103051018@mail.nchu.edu.tw (W.-K.W.)

**Keywords:** palladium complexes, *N*-heterocyclic, indolyl, Suzuki reaction

## Abstract

In this study, a series of *N*-heterocyclic indolyl ligand precursors 2-Py-Py-IndH, 2-Py-Pz-IndH, 2-Py-7-Py-IndH, 2-Py-7-Pz-IndH, and 2-Ox-7-Py-IndH (L^1^H-L^5^H) were prepared. The treatment of ligand precursors with 1 equivalent of palladium acetate affords palladium complexes **1**–**5**. All ligand precursors and palladium complexes were characterized by NMR spectroscopy and elemental analysis. The molecular structures of complexes **3** and **5** were determined by single crystal X-ray diffraction techniques. The application of those palladium complexes **1**–**5** to the Suzuki reaction with aryl halide substrates was examined.

## 1. Introduction

Transition metal-catalyzed cross coupling reactions have been attractive for decades since they are powerful in the formation of various coupling products [1,2,3]. Due to their well-development and broad application in synthetic method, cross coupling was the subject of the Nobel Prize for Chemistry in 2010 [4,5]. Recently, some palladium pincer complexes have been designed and applied in cross coupling reactions [6,7]. This encourages us to develop palladium complexes bearing pincer ligands, which could be applied in cross coupling reactions. Owing to the success in preparation of some metal complexes containing the indole ring system reported by us [8,9] and other groups [10,11,12], introduction of the indole ring system into the pincer ligand precursors will be explored. In this paper, we intended to introduce the *N*-heterocyclic substituents, such as pyridine, pyrazole, or oxazoline as pendant functionalities into the indole ligands in different positions. We hoped the combination of pyridine, pyrazole or oxazoline and indole groups could be the candidates for ligand precursors. The palladium complexes incorporating pyridine-, pyrazole- or oxazoline-indolyl ligands will be reported. Their catalytic activities toward Suzuki reaction are also investigated.

## 2. Results and Discussion

### 2.1. Syntheses and Characterization of Ligand Precursors and Palladium Compounds

In order to prepare the ligand precursors, several bromo-indolyl precursors were synthesized by Fischer-indole synthesis first, followed by Stille reaction (for 2-Py-Py-IndH (L^1^H), 2-Py-7-Py-IndH (L^3^H) and 2-Ox-7-Py-IndH (L^5^H)) or Ullman coupling reaction (for 2-Py-Pz-IndH (L^2^H) and 2-Py-7-Pz-IndH (L^4^H)). The signals of –NH on ^1^H NMR spectra for those indole derivatives were observed around δ 9.36–11.99 ppm. They were characterized by elemental analyses as well. Treatment of these ligand precursors with 1.0 equivalent of Pd(OAc)_2_ in toluene or THF (for **2**) afforded the mono-indolyl palladium acetate complexes **1**–**5**, as shown in Scheme 1.

The disappearance of the N-H signal of functionalized indoles and the appearance of OAc signal are consistent with the proposed structures. The stack ^1^H NMR spectra for L^3^H and [L^3^]Pd(OAc) are shown in Figure 1. 

Compounds **1**–**5** were all characterized by NMR spectroscopy as well as elemental analyses. Suitable crystals of **3** and **5** for structural determination were obtained from CH_2_Cl_2_/hexane solution by the two layers method. The molecular structures are depicted in Figure 2 and Figure 3.

Compounds **3** and **5** demonstrate mono-nuclear form, the bond angles (from 78.0(2)° to 96.0(2)° for **3**, from 79.33(11)° to 97.83(10)° for **5**) around Pd metal centers indicate complexes having a slightly distorted square planar geometry, in which each palladium metal center is coordinated with one indolyl nitrogen atom, two N-heterocyclic nitrogen atoms (two pyridinyl for **3**, one pyridinyl and one oxazolinyl for **5**) and one acetate oxygen atom. Comparisons of some bite angles (°) and bond distances (Å) are given in Table 1. 

The bite angles of five-membered ring N_indolyl_-Pd-N_pyridinyl_ or N_indolyl_-Pd-N_oxazolinyl_ (78.0(2)° for **3** and 79.33(11)° for **5**) are similar to those (81.14(12)° for N_indolyl_-Pd-N_pyridinyl_ [12]; 79.89(7)° or 80.06(9)° for N_imino_-Pd-N_pyridinyl_ [16]; 81.78(8)° or 81.37(11)° for N_imino_-Pd-N_pyridinyl_ [13,18]; 81.39(12)° for N_amido_-Pd-N_quinolinyl_ [19]; 81.76(6)° for N_amido_-Pd-N_pyridinyl_ [14]) found in the literature. The bite angles of six-membered ring N_indolyl_-Pd-N_pyridinyl_ (90.8(8)° for **3** and 89.01(11)° for **5**) are larger than those found in five-membered ring. The bond lengths of Pd-N_indolyl_ (1.941(6)Å for **3** and 1.923(3)Å for **5**) are slightly shorter than those (2.032(3)Å to 2.040(4)Å [11] and 2.012(3)Å [12]) found for some palladium indolyl complexes. This might result from less steric hindrance of ligands in this work. The bond lengths of Pd-N_pyridinyl_ (2.043(4)Å and 2.055(5)Å for **3** and 2.087(3)Å for **5**) are similar to those (2.021(3)Å [12]; 2.1244(17)Å [16]; 2.113(2)Å [18]; 2.119(3)Å [13]) reported in the literature. The bond length of Pd-N_oxazolinyl_ (2.050(3)Å for **3**) is within the range of those (2.0254(14)Å [14]; 2.011(3)Å to 2.052(8)Å [15]; 2.144(3)Å [13] or 1.972(2)Å to 2.055(4)Å [17]) found for some palladium pybox, palladium pendant benzamidinate or palladium anilido-oxazolinate complexes. The bond lengths of Pd-O_OAc_ (2.045(4)Å for **3** and 2.048(2)Å for **5**) are comparable to those (2.035(2)Å [12]; 2.0558(18)Å [16]; 2.0412(18)Å and 2.054(3)Å [20]; 2.036(2)Å [21]; 2.021(4)Å [17]; 2.0545(16)Å [18]; 2.064(3) to 2.045(2)Å [13]) found in the literature.

### 2.2. Catalytic Studies

In our previous work, some palladium complexes bearing different functionalities have been reported and exhibited catalytic activities in cross-coupling reactions [13,16,17,18,20,21]. The palladium complexes discussed above are expected to catalyze the carbon-carbon coupling reactions. For the purpose of comparing reactivity with other corresponding palladium complexes, Suzuki reaction was chosen to demonstrate the catalytic activities. Potential candidates **1**–**5** as catalyst precursors were introduced in the coupling of 4-bromoacetophenone with phenylboronic acid at 70 °C on a 1.0 mol% Pd scale, as shown in Scheme 2. Selected results are listed in Table 2.

The optimized conditions for the reaction were found to be K_2_CO_3_/toluene after several trials with the combination of bases (Cs_2_CO_3_, K_2_CO_3_ and K_3_PO_4_) and solvents (DMSO, DMA, toluene, DMF, THF and EtOH). Higher activities were observed for **3** and **4** with conversion up to 98% and 94%, respectively (entries 1–14). Due to the better activities performed by **3** and **4**, lower concentrations were investigated using 0.5 mol% of catalysts. The reactions gave degrees of conversion to 96% within 1 h at 70 °C for **3**, whereas 53% for **4** (entries 15–16). Complex **3** was tested using 0.5 mol% of the catalyst within 0.5 h, giving a degree of conversion up to 94% (entry 17). Optimized conditions were investigated at room temperature, which gave the degree of conversion to 87% for **3**, and 3% for **4** (entries 18–19). These results demonstrate that the presence of pyridinyl functionalities on 2- and 7-positions of the indole ring shows a better activity in this system for Suzuki coupling reaction. However, poor catalytic activities were observed for the coupling of 4-bromoanisole with phenylboronic acid within 1–2 h (entries 20–21). Complex **3** was tested using more changing substrate 4-chloroacetophenone with phenylboronic acid on 1 mol% Pd scale with K_2_CO_3_/toluene at 70 °C. The reactions exhibited a trace amount of the product within 1–2 h (entries 22–23).

In conclusion, five palladium indolyl complexes bearing *N*-heterocyclic functionalities have been prepared and demonstrated their catalytic activities toward Suzuki C-C coupling reaction. Under optimized conditions, compound **3** exhibits better catalytic activity than compound **4** in catalyzing Suzuki coupling reaction. Based on the results discussed above, aromatic *N*-heterocyclic substituents on 2- and 7-positions of indole ring system exhibit better catalytic activities toward Suzuki C-C coupling reaction. The use of pincer ligands containing a central anionic indolyl fragment outperforms those in which the anionic N-donor is one of the pendant substituents.

## 3. Materials and Methods 

All manipulations were carried out under an atmosphere of dinitrogen using standard Schlenk-line or drybox techniques. Solvents were refluxed over the appropriate drying agent and distilled prior to use. Deuterated solvents were dried over molecular sieves. ^1^H and ^13^C{^1^H} NMR spectra were recorded either on Varian Mercury-400 (400 MHz) or Varian Inova-600 (600 MHz) spectrometers in chloroform-*d* at ambient temperature unless stated otherwise and referenced internally to the residual solvent peak and reported as parts per million relative to tetramethylsilane. Elemental analyses were performed by Elementar Vario ELIV instrument.

2-Acetyl-6-bromopyridine (Ark Pharm, Inc.), phenylhydrazine hydrochloride (Alfa Aesar), (2-bromophenyl)hydrazine hydrochloride (Alfa Aesar), ethyl pyruvate (Alfa Aesar), 2-amino-2-methyl-1-propanol (Fluka), methanesulfonyl chloride (Alfa Aesar), triethylamine (TEDIA), 4-(dimethylamino)pyridine (Alfa Aesar), 2-(tributylstannyl)pyridine (Matrix Scientific), *N*,*N’*-dimethylethylenediamine (DMEDA, Alfa Aesar), 4-bromoacetophenone (Alfa Aesar), 4-bromotoluene (Acros), 4-chloroacetophenone (Acros), *p*-toluenesulfonic acid (TsOH, SHOWA), polyphosphoric acid (PPA, SHOWA), Eaton’s reagent (Alfa Aesar), sodium hydroxide (SHOWA), potassium carbonate (Union Chemical Works), tetrakis(triphenylphosphine)palladium(0) (Aldrich), cesium carbonate (Alfa Aesar), tripotassium phosphate (Alfa Aesar), copper(I) iodide (Aldrich), palladium(II) acetate (Aldrich), phenylboronic acid (Matrix Scientific), and pyrazole(Alfa Aesar) were used as supplied. **2-Py-Br-IndH [22,23]**, **2-Py-7-Br-IndH [24,25]**, and **2-Ox-7-Br-IndH** [26,27,28] were prepared by the literature’s method.

### 3.1. Preparations

**2-Py-Py-IndH****(L^1^H)****.** To a flask containing **2-Py-Br-IndH** (0.54 g, 2.0 mmol) and Pd(PPh_3_)_4_ (0.12 g, 5 mol%), 0.84 mL 2-(tributylstannyl)pyridine (2.6 mmol), and 5 mL toluene were added under nitrogen. The reaction mixture was heated to 110 °C for two days. All volatiles were removed under reduced pressure. The residue was purified by flash column chromatography on silica gel (Ethyl acetate/hexane 1:5). The volatiles were removed under vacuum to give a pale-yellow solid; yield 0.356 g, 66%. ^1^H NMR (400 MHz): δ 7.05(s, 1H, Ar-*H*), 7.13(t, *J =* 7.6 Hz, 1H, Ar-*H*), 7.24(t, *J =* 8.4 Hz, 1H, Ar-*H*), 7.33(t, *J =* 5.6 Hz, 1H, Ar-*H*), 7.45(d, *J =* 8.0 Hz, 1H, Ar-*H*), 7.66(d, *J =* 8.0 Hz, 1H, Ar-*H*), 7.78–7.87(overlap, 3H, Ar-*H*), 8.27(d, *J =* 7.6 Hz, 1H, Ar-*H*), 8.53(d, *J =* 8.0 Hz, 1H, Ar-*H*), 8.71(d, *J =* 4.8 Hz, 1H, Ar-*H*), 9.66(br, 1H, N*H*). ^13^C{^1^H} NMR (100 MHz): δ 100.8, 111.3, 111.9, 119.1, 119.4, 119.8, 120.1, 121.1, 121.2, 123.2, 123.8, 136.7, 136.8, 137.5, 149.2(Ar-*C*H), 129.2, 136.4, 149.6, 155.3, 155.8(*tert*-C). Anal. Calc. for C_18_H_13_N_3_: C, 79.68; H, 4.83; N, 15.49. Found: C, 79.58; H, 4.85; N, 15.17.

**2-Py-Pz-IndH****(L^2^H).** To a flask containing **2-Py-Br-IndH** (0.54 g, 2.0 mmol), K_2_CO_3_ (0.27 g, 2.0 mmol), CuI (0.038 g, 10 mol%) and pyrazole (0.15 g, 2.2 mmol), 0.054 mL DMEDA (0.25 mmol) and 5 mL toluene were added under nitrogen. The reaction mixture was heated to 110 °C for five days. After cooling to room temperature, 10 mL ethyl acetate was added and the mixture was washed with deionized water three time. The organic layer was collected and concentrated. The residue was purified by flash column chromatography on silica gel (ethyl acetate/hexane 1:5). The volatiles were removed to give a yellow solid; yield 0.228 g, 44%. ^1^H NMR (400 MHz): δ 6.53(t, *J =* 2.0 Hz, 1H, Ar-*H*), 7.09(d, *J =* 2.0 Hz, 1H, Ar-*H*), 7.14(t, *J =* 8.0 Hz, 1H, Ar-*H*), 7.26(m, 1H, Ar-*H*), 7.48(d, *J =* 8.4 Hz, 1H, Ar-*H*), 7.66–7.70(overlap, 2H, Ar-*H*), 7.78(d, *J =* 1.6 Hz, 1H, Ar-*H*), 7.83–7.88(overlap, 2H, Ar-*H*), 8.71(d, *J =* 2.8 Hz, 1H, Ar-*H*), 9.36(br, 1H, N*H*). ^13^C{^1^H} NMR (150 MHz): δ 101.5, 107.8, 110.6, 111.3, 117.3, 120.4, 121.3, 123.5, 126.9, 139.3, 142.2(Ar-*C*H), 129.0, 135.8, 136.4, 149.0, 151.0(*tert*-C). Anal. Calc. for C_16_H_12_N_4_: C, 73.83; H, 4.65; N, 21.52. Found: C, 73.76; H, 4.43; N, 21.23.

**2-Py-7-Py-IndH****(L^3^H).** To a flask containing **2-Py-7-Br-IndH** (0.54 g, 2.0 mmol) and Pd(PPh_3_)_4_ (0.12 g, 5 mol%), 0.84 mL 2-(tributylstannyl)pyridine (2.6 mmol), and 5 mL toluene were added under nitrogen. The reaction mixture was heated to 110 °C for two days. All volatiles were removed under reduced pressure. The residue was purified by flash column chromatography on silica gel (ethyl acetate/hexane 1:20). The volatiles were removed under vacuum to give a pale-yellow solid; yield 0.35 g, 65%. ^1^H NMR (400 MHz): δ 7.06(d, *J =* 2.0 Hz, 1H, Ar-*H*), 7.13(m, 1H, Ar-*H*), 7.17–7.22(overlap, 2H, Ar-*H*), 7.65(m, 1H, Ar-*H*), 7.71–7.79(overlap, 4H, Ar-*H*), 7.96(d, *J =* 8.0 Hz, 1H, Ar-*H*), 8.64(d, *J =* 4.8 Hz, 1H, Ar-*H*), 8.83(d, *J =* 4.4 Hz, 1H, Ar-*H*), 11.99(br, 1H, N*H*). ^13^C{^1^H} NMR (100 MHz): δ 100.2, 119.8, 119.9, 119.9, 120.5, 121.2, 121.8, 122.6, 136.3, 136.5, 149.0, 149.4(Ar-*C*H), 121.4, 130.5, 135.1, 137.6, 150.6, 157.5(*tert*-C). Anal. Calc. for C_18_H_13_N_3_: C, 79.68; H, 4.83; N, 15.49. Found: C, 79.41; H, 4.43; N, 15.14.

**2-Py-7-Pz-IndH****(L^4^H).** To a flask containing **2-Py-7-Br-IndH** (0.54 g, 2.0 mmol), K_2_CO_3_ (0.27 g, 2.0 mmol), CuI (0.038 g, 10 mol%) and pyrazole (0.15 g, 2.2 mmol), 0.054 mL DMEDA (0.25 mmol) and 5 mL toluene were added under nitrogen. The reaction mixture was heated to 110 °C for 10 days. After cooling to room temperature, 10 mL ethyl acetate was added and the mixture was washed with deionized water three time. The organic layer was collected and concentrated. The residue was purified by flash column chromatography on silica gel (ethyl acetate/hexane 1:5). The volatiles were removed to give a yellow solid; yield 0.260 g, 50%. ^1^H NMR (400 MHz): δ 6.51–6.52(overlap, 1H, Ar-*H*), 7.06(d, *J =* 2.4 Hz, 1H, Ar-*H*), 7.12(d, *J =* 7.6 Hz, 1H, Ar-*H*), 7.15–7.19(overlap, 1H, Ar-*H*), 7.29(d, *J =* 7.6 Hz, 1H, Ar-*H*), 7.57(d, *J =* 8.0 Hz, 1H, Ar-*H*), 7.71(m, 1H, Ar-*H*), 7.80(d, *J =* 8.0 Hz, 1H, Ar-*H*), 7.89(m, 1H, Ar-*H*), 8.11(d, *J =* 2.8 Hz, 1H, Ar-*H*), 8.65(m, 1H, Ar-*H*), 11.15(br, 1H, N*H*). ^13^C{^1^H} NMR (100 MHz): δ 100.4, 106.8, 110.5, 119.1, 119.7, 119.9, 122.1, 126.7, 136.3, 140.6, 149.3(Ar-*C*H), 125.3, 128.1, 131.7, 137.9, 150.1(*tert*-C). Anal. Calc. for C_16_H_12_N_4_: C, 73.83; H, 4.65; N, 21.52. Found: C, 73.95; H, 4.53; N, 21.27.

**2-Ox-7-Py-IndH****(L^5^H).** To a flask containing **2-Ox-7-Br-IndH** (0.58 g, 2.0 mmol) and Pd(PPh_3_)_4_ (0.12 g, 5 mol%), 0.84 mL 2-(tributylstannyl)pyridine (2.6 mmol) and 5 mL toluene were added under nitrogen. The reaction mixture was heated to 110 °C for two days. All volatiles were removed under reduced pressure. The residue was purified by flash column chromatography on silica gel (ethyl acetate/hexane 1:3). The volatiles were removed under vacuum to give a pale-yellow solid; yield 0.24 g, 41%. ^1^H NMR (400 MHz): δ 1.42(s, 6H, C*H*_3_), 4.13(s, 2H, C*H*_2_), 7.11(d, *J =* 2.0 Hz, 1H, Ar-*H*), 7.18–7.25(overlap, 2H, Ar-*H*), 7.72–7.75(overlap, 2H, Ar-*H*), 7.77–7.81(overlap, 1H, Ar-*H*), 7.95(d, *J =* 8.0 Hz, 1H, Ar-*H*), 8.71(m, 1H, Ar-*H*), 11.56(br, 1H, N*H*). ^13^C{^1^H} NMR (100 MHz): δ 28.4(*C*H_3_), 79.0(*C*H_2_), 105.6, 119.8, 120.1, 121.4, 121.5, 123.3, 136.5, 148.9(Ar-*C*H), 67.8, 121.5, 126.2, 129.2, 135.3, 156.7, 157.2 (*tert*-C). Anal. Calc. for C_18_H_17_N_3_O: C, 74.2; H, 5.88; N, 14.42. Found: C, 73.70; H, 5.88; N, 14.15.

**[****L^1^]Pd(OAc) (1).** To a flask containing **L^1^H** (0.27 g, 1.0 mmol) and Pd(OAc)_2_ (0.22 g, 1.0 mmol), 20 mL toluene was added at room temperature. The reaction mixture was heated to 80 °C for 16 h. The resulting mixture was filtered and the precipitate was collected to afford a reddish solid. Yield, 0.28 g, 65%. ^1^H NMR (600 MHz, DMSO-*d*_6_): δ 2.11(s, 3H, C*H*_3_), 6.81(t, *J =* 6.6 Hz, 1H, Ar-*H*), 6.97(t, *J =* 7.8 Hz, 1H, Ar-*H*), 7.15(s, 1H, Ar-*H*), 7.18(d, *J =* 7.8 Hz, 1H, Ar-*H*), 7.45(d, *J =* 7.8 Hz, 1H, Ar-*H*), 7.82(t, *J =* 6.0 Hz, 1H, Ar-*H*), 7.99(d, *J =* 7.8 Hz, 1H, Ar-*H*), 8.12(d, *J =* 7.8 Hz, 1H, Ar-*H*), 8.18(t, *J =* 8.4 Hz, 1H, Ar-*H*), 8.31(d, *J =* 5.4 Hz, 1H, Ar-*H*), 8.36(m, 1H, Ar-*H*), 8.52(d, *J =* 8.4 Hz, 1H, Ar-*H*). ^13^C{^1^H} NMR (150 MHz, DMSO-*d*_6_): δ 23.4(O-C(=O)*C*H_3_), 104.4, 114.2, 117.9, 118.5, 120.2, 121.2, 122.9, 123.9, 127.5, 140.6, 140.9, 150.0(Ar-*C*H), 109.5, 128.3, 145.8, 146.1, 153.4, 155.4, 156.8, 176.0(*tert*-C). Anal. Calc. for C_20_H_15_N_3_O_2_Pd: C, 55.12; H, 3.47; N, 9.64. Found: C, 53.12; H, 3.30; N, 9.00.

**[****L^2^]Pd(OAc) (2).** To a flask containing **L^2^H** (0.26 g, 1.0 mmol) and Pd(OAc)_2_ (0.22 g, 1.0 mmol), 20 mL THF was added at room temperature. The reaction mixture was heated to 60 °C for 16 h. The resulting mixture was filtered and the precipitate was collected to afford a dark-green solid. Yield, 0.28 g, 66%. ^1^H NMR (600 MHz, DMSO-*d*_6_): δ 2.00(s, 3H, C*H*_3_), 6.77(m, 1H, Ar-*H*), 6.86(m, 1H, Ar-*H*), 6.92(m, 1H, Ar-*H*), 7.11(m, 1H, Ar-*H*), 7.34(m, 1H, Ar-*H*), 7.40(d, *J =* 7.8 Hz, 1H, Ar-*H*), 7.74(m, 1H, Ar-*H*), 7.79(m, 1H, Ar-*H*), 8.02(d, *J =* 1.8 Hz, 1H, Ar-*H*), 8.17(t, *J =* 7.8 Hz, 1H, Ar-*H*), 9.04(d, *J =* 3.0 Hz, 1H, Ar-*H*). ^13^C{^1^H} NMR (150 MHz, DMSO-*d*_6_): δ 23.0(O-C(=O)*C*H_3_), 104.2, 106.4, 109.9, 114.1, 116.3, 118.1, 121.2, 122.9, 131.7, 142.5, 145.5(Ar-*C*H), 128.2, 145.3, 145.9, 147.2, 154.1, 176.0(*tert*-C). Anal. Calc. for C_18_H_14_N_4_O_2_Pd: C, 50.90; H, 3.32; N, 13.19. Found: C, 48.78; H, 3.10; N, 12.47.

**[****L^3^]Pd(OAc) (3).** To a flask containing **L^3^H** (0.27 g, 1.0 mmol) and Pd(OAc)_2_ (0.22 g, 1.0 mmol), 20 mL toluene was added at room temperature. The reaction mixture was heated to 80 °C for 16 h. The resulting mixture was filtered and the precipitate was collected to afford a yellow solid. Yield, 0.26 g, 60%. ^1^H NMR (600 MHz): δ 2.26(s, 3H, C*H*_3_), 6.90(s, 1H, Ar-*H*), 7.05(t, *J =* 7.8 Hz, 1H, Ar-*H*), 7.10(m, 1H, Ar-*H*), 7.16(m, 1H, Ar-*H*), 7.69(d, *J =* 7.8 Hz, 2H, Ar-*H*), 7.76–7.79(overlap, 2H, Ar-*H*), 7.86(m, 1H, Ar-*H*), 8.15(d, *J =* 5.4 Hz, 1H, Ar-*H*), 8.33(m, 1H, Ar-*H*), 8.88(m, 1H, Ar-*H*). ^13^C{^1^H} NMR (150 MHz): δ 24.7(O-C(=O)*C*H_3_), 101.7, 119.1, 119.4, 121.5, 121.6, 121.9, 122.4, 125.9, 137.8, 138.9, 149.8, 152.3 (Ar-*C*H), 119.8, 129.1, 134.8, 144.9, 151.2, 156.6, 177.9(*tert*-C). Anal. Calc. for C_20_H_15_N_3_O_2_Pd: C, 55.12; H, 3.47; N, 9.64. Found: C, 54.55; H, 3.13; N, 9.92.

**[****L^4^]Pd(OAc) (4).** To a flask containing **L^4^H** (0.26 g, 1.0 mmol) and Pd(OAc)_2_ (0.22 g, 1.0 mmol), 20 mL toluene was added at room temperature. The reaction mixture was heated to 80 °C for 16 h. The resulting mixture was filtered and the precipitate was collected to yellow solid. Yield, 0.21 g, 50%. ^1^H NMR (600 MHz): δ 2.24(s, 3H, C*H*_3_), 6.57(t, *J =* 3.0 Hz, 1H, Ar-*H*), 6.87(s, 1H, Ar-*H*), 6.95(t, *J =* 8.4 Hz, 1H, Ar-*H*), 7.12(m, 1H, Ar-*H*), 7.19(d, *J =* 7.8 Hz, 1H, Ar-*H*), 7.50(d, *J =* 7.8 Hz, 1H, Ar-*H*), 7.69(m, 1H, Ar-*H*), 7.81(m, 1H, Ar-*H*), 7.87(m, 1H, Ar-*H*), 8.17(m, 1H, Ar-*H*), 8.40(m, 1H, Ar-*H*). ^13^C{^1^H} NMR (150 MHz): δ 24.1(O-C(=O)*C*H_3_), 100.9, 107.9, 108.5, 118.2, 119.7, 121.4, 121.6, 128.1, 139.1, 142.8, 150.1(Ar-*C*H), 124.4, 129.3, 130.5, 144.9, 156.5, 178.1(*tert*-C). Anal. Calc. for C_18_H_14_N_4_O_2_Pd: C, 50.90; H, 3.32; N, 13.19. Found: C, 50.91; H, 3.37; N, 12.93.

**[****L^5^]Pd(OAc) (5).** To a flask containing **L^5^H** (0.29 g, 1.0 mmol) and Pd(OAc)_2_ (0.22 g, 1.0 mmol), 20 mL toluene was added at room temperature. The reaction mixture was heated to 80 °C for 16 h. The resulting mixture was pumped to dryness. The residue was washed with 20 mL hexane to afford a yellow solid. Yield, 0.27 g, 60%. ^1^H NMR (600 MHz): δ 1.50(s, 6H, C*H*_3_), 2.15(s, 3H, C*H*_3_), 4.50(s, 2H, C*H*_2_), 6.86(s, 1H, Ar-*H*), 7.06(t, *J =* 7.8 Hz, 1H, Ar-*H*), 7.15(m, 1H, Ar-*H*), 7.70(d, *J =* 8.4 Hz, 1H, Ar-*H*), 7.78(d, *J =* 7.8 Hz, 1H, Ar-*H*), 7.84(m, 1H, Ar-*H*), 8.27(d, *J =* 7.8 Hz, 1H, Ar-*H*), 8.71(d, *J =* 6.0 Hz, 1H, Ar-*H*). ^13^C{^1^H} NMR (150 MHz): δ 24.3(O-C(=O)*C*H_3_), 27.0(*C*H_3_), 83.5(*C*H_2_), 105.0, 119.4, 121.9, 122.4, 122.9, 126.9, 137.9, 151.8(Ar-*C*H), 64.5, 119.9, 128.4, 131.0, 135.3, 150.9, 168.4, 178.1(*tert*-C). Anal. Calc. for C_20_H_19_N_3_O_3_Pd: C, 52.70; H, 4.20; N, 9.22. Found: C, 52.54; H, 4.71; N, 8.91.

General procedure for the Suzuki-type coupling reaction: A prescribed amount of catalyst, aryl halide (1 equiv), phenylboronic acid (1.5 equiv), and base (2 equiv) was placed in a Schlenk tube under nitrogen. The solvent (2 mL) was added by syringe, and the reaction mixture was heated to the prescribed temperature for the prescribed time. A small portion of the resulting mixture was taken and pumped to dryness. The residue was dissolved in ethyl acetate and passed through a short silica gel column. The ^1^H NMR spectrum of filtrate was taken after removal of the solvent. Conversions were determined by the integral intensities between substrates and products on the ^1^H NMR spectra.

### 3.2. Crystal Structure Data

Crystals were grown from CH_2_Cl_2_/hexane solution (**3** or **5**) by the two layers method and isolated by filtration. Suitable crystals of **3** or **5** were mounted onto glass fiber using perfluoropolyether oil and cooled rapidly in a stream of cold nitrogen gas to collect diffraction data at 150 K using Bruker APEX2 diffractometer, and intensity data were collected with ω scans. The data collection and reduction were performed with the SAINT software [29] and the absorptions were corrected by SADABS [30]. The space group determination was based on a check of the Laue symmetry and systematic absences, and was confirmed using the structure solution. The structure was solved and refined with SHELXTL package [31]. All non-H atoms were located from successive Fourier maps, and hydrogen atoms were treated as a riding model on their parent C atoms. Anisotropic thermal parameters were used for all non-H atoms, and fixed isotropic parameters were used for H-atoms. A drawing of the molecular structure was done by using Oak Ridge Thermal Ellipdoid Plots (ORTEP) [32]. Some details of the data collection and refinement are given in Table 3. Both compounds are disordered. One restraint has been done on the C9 and C13 of compound **3**.

## Data Availability

Crystallographic data for the structure in this paper have been deposited with the Cambridge Crystallographic Data Centre as supplementary publication numbers, CCDC no. 2092418–2092419 for compounds **3** and **5**. These detail crystal data can be obtained free of charge from The Cambridge Crystallographic Data Centre via www.ccdc.cam.ac.uk/data_request/cif (access date 2021/06/26).

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
