# Peer review of "Synthesis, Characterization, and Catalytic Application of Palladium Complexes Containing Indolyl-NNN-Type Ligands"

_molecules, 2021, doi:10.3390/molecules26154426_

Round 1

Reviewer 1 Report

The manuscript of Lo et al. contains two crystal structures of palladium compounds 3 and 5. I was mainly looking at these structure determinations.

According to the experimental section, both crystal structures were measured at a temperature of 150 K. Surprisingly, the anisotropic displacement parameters in compound 3 are more than twice as large as in compound 5. Therefore I am convinced that the measurement temperature is different for the two cases. The authors are requested to check carefully.

In addition to this major issue I have some minor points:

1) The figure captions of Figure 1 and Figure 2 should state the probability level of the ellipsoids.

2) In section 2.1 a large number of geometrical values in compared. In my opinion this is quite difficult to read. It is probably better for the comparison to have the numerical values in the form of a table.

3) There is a discrepancy between the software which is described in the section "Crystal structure data" (page 9) and the deposited cif-files. According to the cif-files, the integration was done with SAINT and the absorption correction with SADABS. This should be made consistent (and the literature references as well).

4) The manuscript says that the crystals were obtained from dichloromethane/hexane. It is not clear if this is vapour diffusion, liquid diffusion or cooling (or something else).

5) In the pdf-file which I obtained for review, in Table 3 and the footnotes of Table 3, all Greek symbols are missing (alpha, beta, gamma, rho, sigma).

6) In the footnote of Table 3 it is stated that w=0.1. This is inconsistent with the deposited cif-files which show different weighting schemes.

7) In the experimental section of the crystal structures, all used restraints should be explained. If an extinction correction was applied, this should also be mentioned.

Author Response

The manuscript of Lo et al. contains two crystal structures of palladium compounds 3 and 5. I was mainly looking at these structure determinations.

According to the experimental section, both crystal structures were measured at a temperature of 150 K. Surprisingly, the anisotropic displacement parameters in compound 3 are more than twice as large as in compound 5. Therefore I am convinced that the measurement temperature is different for the two cases. The authors are requested to check carefully.

Response : I checked the data with the operator. Both crystal structures were measured at a temperature of 150 K. 

In addition to this major issue I have some minor points:

1) The figure captions of Figure 1 and Figure 2 should state the probability level of the ellipsoids.

Response 1: The probability levels of the ellipsoids have been stated in both Figures. The changes have been highlighted.

2) In section 2.1 a large number of geometrical values in compared. In my opinion this is quite difficult to read. It is probably better for the comparison to have the numerical values in the form of a table.

Response 2: The Table has been created as Table 1. Therefore, the Table1 in the original manuscript should be changed to Table 2.

3) There is a discrepancy between the software which is described in the section "Crystal structure data" (page 9) and the deposited cif-files. According to the cif-files, the integration was done with SAINT and the absorption correction with SADABS. This should be made consistent (and the literature references as well).

Response 3: The programs have been corrected in the section of "Crystal structure data". The literature references have been changed as well. These changes have been highlighted.

4) The manuscript says that the crystals were obtained from dichloromethane/hexane. It is not clear if this is vapour diffusion, liquid diffusion or cooling (or something else).

Response 4: The statement ‘by the two layers method’ has been added in the section of "Crystal structure data". The changes have been highlighted.

5) In the pdf-file which I obtained for review, in Table 3 and the footnotes of Table 3, all Greek symbols are missing (alpha, beta, gamma, rho, sigma).

Response 5: The Greek symbols have been corrected in Table3 and the footnotes of Table3. The Greek symbols (δ) in the ‘Preparations’ section has been corrected as well. These changes have been highlighted.

6) In the footnote of Table 3 it is stated that w=0.1. This is inconsistent with the deposited cif-files which show different weighting schemes.

Response 6: We removed “w=0.1” in the weighting scheme and rechecked the cif files. The “wR2” values were revised in highlight marker in the manuscript.

7) In the experimental section of the crystal structures, all used restraints should be explained. If an extinction correction was applied, this should also be mentioned.

Response 7: The sentence ’ Both compounds are disordered. One restraint has been done on C9 and C13 of compound 3.’ has been added in the experimental section of the crystal structures. The sentence has been highlighted.

Reviewer 2 Report

The work presented by Chen and collaborators presents a new family of indolyl-NNN terdentate ligands and their corresponding Pd-acetate complexes. The latter compounds have been tested as catalysts in two different cross-coupling benchmark processes. The ligands introduce an interesting diversity in terms of the nature of the central donor and the pendant functionalities, which could open interesting avenues for further studies. Unfortunately, in the current version of the manuscript, the authors fail to present and analyze this potential in a clear and systematic manner. Additionally, the experimental design of the catalytic studies lacks the required systematic structure to support the conclusions claimed.

In this reviewer’s opinion, some aspects of the manuscript should be improved before being considered acceptable for publication. Major changes that should be addressed include:

  • Description of the NMR of both ligands and complexes. The current description of the NMR spectra of the ligands, just mentioning the presence of N-H signals at the low field region of the spectra, and their disappearance upon metal coordination is rather poor. A more accurate description (or even a Figure containing a stack plot of the 1H NMR spectra) could give support to the claimed purity of both ligands and complexes.
  • Design of catalytic experiments:
    • Suzuki reaction: The five catalytic systems should be tested under identical reaction conditions to claim which is the best catalyst for the process. Catalyst 1 and 2 have been tested using catalytic conditions that are not comparable with the rest of the series.
    • Heck reaction:  catalyst optimization has been performed using a substrate that gives full conversions at nearly all the conditions assayed. With this set of experiments it is not possible to assess which are the optimal conditions for the process. To get conclusive results the optimization should be performed at lower temperatures, shorter reaction times, lower catalyst loadings or using a more challenging substrate. If the authors do not want to add experimental work, consider at least rephrasing the description of the experiments.

Other aspects that could help to improve the manuscript are:

-An initial Scheme including the explicit drawing of all the ligands would facilitate understanding the ligands design and their comparison. Additionally, I would suggest simplifying the labeling of the ligands to L1-L5.  

-Include in the experimental part the description of the procedure used to determine conversions in the catalytic experiments. 1H NMR analysis of aliquots, evaporated samples or purified samples?

-Authors claim that the catalytic processes selected were intended for comparative purposes. Therefore, a clear comparison of the catalytic activity with that of other related systems should be included in the discussion. The current comment in the manuscript is too vague “In some cases, compound 4 shows even better catalytic activities than those reported in the literature for catalyzing Heck coupling reaction” Could the authors be more specific?

-Do the authors have an explanation for the slightly better catalytic activity obtained in the Suzuki reaction when catalyst 3 was used with a lower catalyst loading (entries 3 vs 13 in table 1)?

-I suggest rephrasing the conclusions to a more conceptual description of the type “the use of pincer ligands containing a central anionic indolyl fragment outperforms those in which the anionic N-donor is one of the pendant substituents”.

-Page 5 line 120. Consider rephrasing:  “Compound 4 was optimized as better catalyst using bromobenzene as substrate…2

Author Response

The work presented by Chen and collaborators presents a new family of indolyl-NNN terdentate ligands and their corresponding Pd-acetate complexes. The latter compounds have been tested as catalysts in two different cross-coupling benchmark processes. The ligands introduce an interesting diversity in terms of the nature of the central donor and the pendant functionalities, which could open interesting avenues for further studies. Unfortunately, in the current version of the manuscript, the authors fail to present and analyze this potential in a clear and systematic manner. Additionally, the experimental design of the catalytic studies lacks the required systematic structure to support the conclusions claimed.

In this reviewer’s opinion, some aspects of the manuscript should be improved before being considered acceptable for publication. Major changes that should be addressed include:

  • Description of the NMR of both ligands and complexes. The current description of the NMR spectra of the ligands, just mentioning the presence of N-H signals at the low field region of the spectra, and their disappearance upon metal coordination is rather poor. A more accurate description (or even a Figure containing a stack plot of the 1H NMR spectra) could give support to the claimed purity of both ligands and complexes.

Response 1: A stack 1H NMR spectra for L3H and [L3]Pd(OAc) has been created as Figure1. Therefore, the figure’s no. should be changed for the followings. The statement ‘the appearance of OAc signal are consistent with the proposed structures. The stack 1H NMR spectra for L3H and [L3]Pd(OAc) are shown in Figure 1’ has been added. These changes have been highlighted in the manuscript. 

  • Design of catalytic experiments:
    • Suzuki reaction: The five catalytic systems should be tested under identical reaction conditions to claim which is the best catalyst for the process. Catalyst 1 and 2 have been tested using catalytic conditions that are not comparable with the rest of the series.
  • Response 2: Two entries using the optimized conditions have been done and added in catalytic table (Table 2 entries 3-4). 
    • Heck reaction:  catalyst optimization has been performed using a substrate that gives full conversions at nearly all the conditions assayed. With this set of experiments it is not possible to assess which are the optimal conditions for the process. To get conclusive results the optimization should be performed at lower temperatures, shorter reaction times, lower catalyst loadings or using a more challenging substrate. If the authors do not want to add experimental work, consider at least rephrasing the description of the experiments.
    • Response 3: Please see the Corresponding changes for Heck-type coupling reaction in Response 5.

Other aspects that could help to improve the manuscript are:

-An initial Scheme including the explicit drawing of all the ligands would facilitate understanding the ligands design and their comparison. Additionally, I would suggest simplifying the labeling of the ligands to L1-L5.  

Response 4: The changes have been added in the Abstract, Scheme and Preparations. These changes have been highlighted in the revised manuscript.

-Include in the experimental part the description of the procedure used to determine conversions in the catalytic experiments. 1H NMR analysis of aliquots, evaporated samples or purified samples?

Response 5: 

For Suzuki-type coupling reaction:

The sentences ‘A small portion of resulting mixture was taken and pumped to dryness. The residue was dissolved in ethyl acetate and passed through a short silica gel column. The 1H NMR spectrum of filtrate was taken after removal of solvent. Conversions were determined by the integral intensities between substrates and products on the 1H NMR spectra.’ have been added in the experimental part. These changes have been highlighted in the revised manuscript.

For Heck-type coupling reaction:

This question reminds us to reconsider if it’s adequate to demonstrate catalytic activities in conversion using the liquid substrate. This might result in deviation if the boiling point of liquid substrate is just a little bit higher than that of solvent. Therefore, we intend to remove Scheme 3 and Table 2 in the revised manuscript. The discuss for Heck-type coupling reaction in the context and the General procedure for the Heck-type coupling reaction in the experimental part are removed as well.

Based on the results, the catalytic behavior of these compounds in Suzuki coupling reaction still match the discussion and conclusion in the context.

The Graphic abstract will be re-submitted as well.

-Authors claim that the catalytic processes selected were intended for comparative purposes. Therefore, a clear comparison of the catalytic activity with that of other related systems should be included in the discussion. The current comment in the manuscript is too vague “In some cases, compound 4 shows even better catalytic activities than those reported in the literature for catalyzing Heck coupling reaction” Could the authors be more specific?

Response 6: Please see the Corresponding changes for Heck-type coupling reaction in Response 5.

-Do the authors have an explanation for the slightly better catalytic activity obtained in the Suzuki reaction when catalyst 3 was used with a lower catalyst loading (entries 3 vs 13 in table 1)?

Response 7: The catalytic activities have been re-confirmed using freshly-made catalysts. The catalytic results have been corrected and the isolated yields have been reported in parentheses. Several entries have been done and added in catalytic table (Table 2 in revised manuscript).

-I suggest rephrasing the conclusions to a more conceptual description of the type “the use of pincer ligands containing a central anionic indolyl fragment outperforms those in which the anionic N-donor is one of the pendant substituents”.

Response 8: The sentence has been added in the conclusion.

-Page 5 line 120. Consider rephrasing:  “Compound 4 was optimized as better catalyst using bromobenzene as substrate…2

Response 9: Please see the Corresponding changes for Heck-type coupling reaction in Response 5. 

Round 2

Reviewer 2 Report

I appreciate the author's effort to respond to all the questions and improve the manuscript according to the suggestions. I am satisfied with the current version. It still requires some minor language/grammar checks, but I do support its publication.

Author Response

I appreciate the author's effort to respond to all the questions and improve the manuscript according to the suggestions. I am satisfied with the current version. It still requires some minor language/grammar checks, but I do support its publication.

Response: Thanks for reviewer's suggestions and help.

Some language/grammar checks have been done by us following Academic Editor Notes. We hope the reviewer can be satisfied by the corresponding changes.